# Auditing the impact of social media's policy shift on anti-vaccine discourse: A large language model-driven empirical study

Yufei Li[1,☯], Tianhao Chen[1,☯], Yanjie Zhao[2,3], Wei Ke[1], Patrick Pang [1,*], Dana McKay [4], Shanton Chang[5], Nancy Baxter[6]

1 Faculty of Applied Sciences, Macao Polytechnic University, Macao SAR, China, 2 Beijing Key Laboratory of Mental Disorders, National Clinical Research Center for Mental Disorders & National Center for Mental Disorders, Beijing Anding Hospital, Capital Medical University, Beijing, China, 3 Advanced Innovation Center for Human Brain Protection, Capital Medical University, Beijing, China, 4 School of Computing Technologies, RMIT University, Melbourne, Australia, 5 School of Computing and Information Systems, The University of Melbourne, Melbourne, Australia, 6 Faculty of Medicine and Health, The University of Sydney, Sydney, Australia

☯ These authors contributed equally to this work.
* mail@patrickpang.net

## Abstract

The sudden termination of X's (formerly Twitter) misinformation policy on November 23, 2022, provides an opportunity to assess the effects of lifting content moderation restrictions on vaccine-related discourse. This study examines changes in the prevalence, thematic composition, and engagement of anti-vaccine discourse following X's policy shift, analyzing tweets from a seven-day period before and after the policy termination (November 16–30, 2022), excluding the announcement date itself from regression analyses. Using GPT-4o for stance classification, thematic categorization, and stance consistency assessment, with validation through external benchmarks and cross-annotator agreement, we find that anti-vaccine tweets increased significantly post-policy (OR = 1.60, 95% CI: 1.50–1.72), particularly via retweets, suggesting content amplification. Sensitivity analyses excluding highly retweeted content revealed that the policy change was also associated with increased creation of new anti-vaccine content. Thematically, health concerns over vaccination became more prominent, while conspiracy-related and anti-mandate narratives declined in relative prevalence. Stance consistency in quote tweets increased, indicating reinforced ideological alignment in anti-vaccine discourse. These results suggest that content moderation policies may constrain both the volume and amplification of anti-vaccine content, with policy removal associated with rapid shifts in discourse patterns.

## Introduction

The spread of anti-vaccine discourse on social media plays a crucial role in shaping public perceptions of vaccines and influencing vaccine hesitancy [1,2]. Previous

**Data availability statement:** https://doi.org/10.5281/zenodo.19019934.

**Funding:** The author(s) received no specific funding for this work.

**Competing interests:** The authors have declared that no competing interests exist.

studies have shown that exposure to vaccine misinformation can significantly reduce willingness to get vaccinated [3]. A large share of the misinformation circulating during the pandemic centered on vaccines, spreading doubts about their safety and effectiveness. These narratives contributed directly to growing vaccine hesitancy and public distrust in health guidance. [4,5]. Given the widespread impact of online discussions on public health behaviors [3,6,7], understanding the dynamics of anti-vaccine discourse in response to platform policy changes is of critical importance [8].

Social media platforms have long grappled with the challenge of balancing content moderation and freedom of speech when addressing health misinformation [9]. In response to the pandemic, X implemented a misinformation enforcement policy in 2020 [10,11], introducing content labeling, warning systems, and deplatforming measures to combat misleading claims about COVID-19 vaccines. Similar efforts were undertaken by Facebook [12], Instagram [13], and YouTube [14], which enacted policies to remove or restrict misleading content that contradicted public health guidance [15]. These measures were designed to curb the spread of misinformation and promote credible health information.

However, in November 2022, X, under Elon Musk's leadership, abruptly terminated its COVID-19 misinformation enforcement policy [16]. Musk, describing himself as a "free speech absolutist," argued that social media platforms should not remove content that is offensive but still legal, stating that if something is in a "gray area," it should be allowed to remain [17,18]. While extensive research has examined the *implementation* of content moderation policies and their effectiveness in reducing misinformation spread, far less is known about the consequences of *removing* such policies once established. The effects of policy removal may not simply mirror the effects of policy implementation in reverse; understanding these dynamics is crucial for evidence-based platform governance. This policy reversal provides an opportunity to examine how the announcement of policy termination was associated with changes in the prevalence and nature of anti-vaccine discourse on the platform.

In this study, we investigate the short-term impact of X's policy termination on anti-vaccine discourse. Specifically, we seek to answer the following research questions:

- **RQ1**: How does the termination of X's COVID-19 vaccine misinformation policy impact the volume of anti-vaccine tweets?

- **RQ2**: Does the policy change shift the thematic composition of anti-vaccine discourse, and if so, which narratives became more or less prominent?

- **RQ3**: How does the policy termination affect the stance consistency of quote tweets within vaccine-related discussions?

To address these questions, we collected a dataset of COVID-19 vaccine-related tweets from November 16–30, 2022. For regression analyses, we compared seven days before (November 16–22) and seven days after (November 24–30) the policy announcement, excluding the announcement date itself (November 23) to isolate pre- and post-announcement periods. Using GPT-4o as an annotation tool, we

categorized tweets based on their vaccine stance (pro-vaccine, anti-vaccine, neutral) and assigned the identified anti-vaccine tweets into eight thematic categories [8]. We validated the LLM classifier through external benchmark testing and cross-annotator agreement assessment. We then applied logistic regression models to analyze the changes in anti-vaccine prevalence and topic distribution following the policy termination, with sensitivity analyses to assess the robustness of findings when excluding highly retweeted content.

This study provides empirical insights into the role of content moderation in shaping online vaccine discourse. By leveraging a real-world policy shift as a pre-post observational design, we document changes in anti-vaccine discourse following X's policy termination. In our analysis, we examine three key dimensions: (1) *amplification*, the degree to which anti-vaccine content spreads through retweets; (2) *thematic composition*, the specific narratives expressed in anti-vaccine discourse; and (3) *stance consistency*, whether users quoting vaccine-related tweets express agreement or disagreement with the original content. Our findings have implications for platform regulation, misinformation governance, and the broader intersection of free speech and public health policies.

## Related work

### Anti-vaccine discourse on social media

Social media has become a major battleground for vaccine-related discourse, with anti-vaccine rhetoric proliferating across platforms. While pro-vaccine content generally dominates public health messaging, anti-vaccine narratives have been found to engage audiences more effectively due to their emotive and polarizing nature [19]. Research has identified several recurring themes within anti-vaccine discourse, including concerns about vaccine safety, distrust of pharmaceutical companies and government institutions, and the belief in personal health autonomy [20,21]. In addition, conspiracy theories, such as claims that vaccines are used for population control or contain microchips, have played a substantial role in fueling skepticism, particularly during the COVID-19 pandemic [22,23]. Despite lacking scientific support, these narratives have gained traction through social media mechanisms that facilitate their amplification within vaccine-hesitant communities.

Recent studies indicate that vaccine misinformation does not need to be factually incorrect to influence hesitancy. Allen et al. [24] found that the most influential content was not outright falsehoods but rather vaccine-skeptical narratives that raised concerns without making verifiably false claims. Broniatowski et al. [25] similarly found that COVID-19 misinformation was less likely to spread than accurate content, suggesting that amplification mechanisms rather than inherent virality drive misinformation visibility.

Social media platforms facilitate the rapid spread of anti-vaccine content through algorithmic amplification and community clustering [26,27]. Studies indicate that engagement-driven ranking systems amplify misleading yet technically accurate narratives [28,29], while automated accounts and coordinated networks systematically amplify skeptical narratives [28]. Hwang et al. [30] identified distinct thematic clusters in vaccine discourse on Twitter, providing a framework for analyzing anti-vaccine narratives.

While much research has analyzed the spread and impact of anti-vaccine discourse, fewer studies have examined how changes in content moderation policies influence the evolution of these narratives. The sudden removal of X's COVID-19 misinformation policy presents an opportunity to assess how the withdrawal of content moderation affects the prevalence, themes, and engagement patterns of anti-vaccine rhetoric.

### Content moderation and policy changes in shaping online discourse

Content moderation policies play a crucial role in controlling the spread of health misinformation. Previous research has demonstrated that interventions such as labeling, fact-checking, and deplatforming can effectively reduce engagement with misinformation [9,31]. For instance, content labeling reduces the perceived credibility of misinformation, while

fact-checking interventions mitigate belief persistence [32,33]. However, the effectiveness of content moderation remains debated. While some scholars argue that removing misleading content reduces public exposure to harmful narratives [15], others suggest that excessive moderation may push misinformation to alternative platforms with weaker oversight [34,35].

X's 2020 COVID-19 misinformation policy, along with similar initiatives by Facebook and YouTube, demonstrated a platform-wide commitment to mitigating public health misinformation [11,12]. However, concerns over freedom of expression have led to ongoing debates about the balance between regulation and free speech [9]. Past research has shown that deplatforming efforts can significantly disrupt the spread of misinformation [36], as seen in X's removal of QAnon-related content, which reduced the influence of key conspiracy theorists [37]. Conversely, platforms with relaxed moderation policies, such as Parler and Gab, have experienced surges in extremist and misleading content [38].

Algorithmic amplification remains a challenge for misinformation control, as platforms struggle to prevent users from being funneled into misinformation pathways [39]. Similarly, research on COVID-19 misinformation has examined how moderation efforts shape discourse trends [8]. Lanier et al. [40] analyzed COVID-19 disinformation hashtags such as #scamdemic and #plandemic on Parler, demonstrating how vaccine-skeptical narratives proliferate on platforms with minimal moderation. However, relatively few studies have analyzed the effects of removing misinformation policies after they have been established. Mitts et al. [41] found that the removal of anti-vaccine groups on Facebook led to increased anti-vaccine rhetoric on X, highlighting the potential for cross-platform spillover effects. Additionally, Broniatowski et al. [42] showed that Facebook's misinformation removal efforts did not always decrease overall engagement with anti-vaccine content, as users adapted to platform restrictions. Hobbs et al. [43] examined the sharing of low-credibility URLs on Twitter during reporting of blood clots following the AstraZeneca vaccine, showing how health scares can trigger spikes in misinformation sharing.

Existing research has primarily focused on the implementation of moderation policies rather than their removal. While numerous studies have examined how introducing content moderation reduces misinformation spread [9,15,31], empirical evidence on policy removal remains scarce. The few exceptions [41,42] focus on specific interventions rather than wholesale policy termination. The sudden termination of X's COVID-19 misinformation enforcement policy provides a pre-post observational design to assess how policy withdrawal was associated with changes in misinformation dynamics. This study contributes to this gap by systematically analyzing the short-term consequences of reversing content moderation efforts, with implications for both platform governance and public health outcomes.

## Materials and methods

### Data collection

To analyze the changes following X's policy shift on anti-vaccine discourse, we utilized an existing dataset of COVID-19-related tweets provided by [44]. Although the dataset description was published in 2021, the data collection was continuously maintained and updated through November 2022, covering the period of our study. This dataset, originally collected through Twitter's Academic API v2, contains tweet IDs that were made publicly available for research purposes. Following X's data access policies, we rehydrated these tweet IDs to obtain the full text of the corresponding tweets.

To specifically focus on vaccine-related discussions, we applied an additional keyword filtering step inspired by [45]. After retrieving the full dataset, we extracted tweets containing vaccine-related keywords such as "mRNA vaccine," "booster shot," and vaccine manufacturer names (for example, "Pfizer" and "Moderna"). We show a few examples in Table 1. This two-step filtering approach ensured that our dataset retained a broad range of COVID-19 discourse while allowing us to analyze vaccine-related narratives in response to X's policy change.

The collected tweets were categorized into three distinct types, as shown in Table 2. Original tweets (TW) are those composed and posted by users, quote tweets (QT) include a reshared tweet with an added comment, and retweeted tweets (RT) are simply reshared without any modifications.

**Table 1. Sample keywords used for collecting COVID-19 and vaccine-related tweets.**

| Keywords used for COVID-19 tweet collection (from [44]) |
| --- |
| coronavirus, COVID-19, pandemic, 2019nCoV, CoronaOutbreak, WuhanVirus. |
| Additional sample vaccine-related keywords (from [45]) |
| covid19vaccine, covidvaccine, coronavirusvaccine, vaccination, covid19 pfizer, pfizercovidvaccine, oxfordvaccine, getvaccinated, moderna, mrna vaccinate, covax, coronavirus moderna, vax. |

**Table 2. Categorization of collected tweets.**

| Tweet type | Description |
| --- | --- |
| Original tweets (TW) | Tweets originally composed and posted by users. |
| Quote tweets (QT) | Tweets that share another tweet with an added comment. |
| Retweeted tweets (RT) | Tweets reshared without any modifications or additional comments. |

To improve data quality, we applied preprocessing steps following [44], including deduplication, language filtering (retaining only English tweets), and removal of URLs and redundant whitespace. After preprocessing, we performed stratified sampling based on tweet type and tweet date to ensure a balanced representation across different categories.

Unlike prior studies that often exclude retweets to focus on original content, we retain retweeted tweets to examine the prevalence of anti-vaccine discourse in response to X's policy shift. This design choice reflects our focus on the information environment to which users are exposed: retweets contribute to the overall concentration of anti-vaccine content in users' feeds, which is relevant to public health outcomes regardless of content originality. The inclusion of retweets allows us to analyze how anti-vaccine narratives propagate within the platform and assess the patterns of content sharing following the removal of content moderation policies. To address potential concerns about highly viral content driving results, we also conduct sensitivity analyses excluding tweets with ≥ 50 retweets.

## Data annotation method

We employed the GPT-4o API for automated tweet annotation, leveraging the Chain-of-Thought prompting technique to enhance the reasoning capabilities of the LLM. Each annotated tweet includes not only a categorical label but also the LLM's reasoning process and confidence score to ensure classification transparency and robustness.

All three classification tasks, vaccine stance classification, anti-vaccine theme categorization, and stance consistency analysis, were conducted using GPT-4o. The LLM was prompted with task-specific instructions and example-based guidance to improve classification performance across different categories.

## Vaccine stance classification

Tweets were categorized into four stance labels: pro-vaccine, anti-vaccine, neutral, and no stance mentioned, as detailed in Table 3. The classification was performed based on the textual content of each tweet, considering explicit statements, implied meanings, and context.

For retweets (RT), we assume that the poster's vaccine stance aligns with the original tweet, based on prior research [46], which indicates that retweeting behavior generally signifies endorsement rather than opposition. For quote tweets (QT), stance determination requires contextual reasoning by analyzing both the quoted tweet and the poster's comment. Given the complexity of vaccine discourse, we utilize GPT-4o's advanced language comprehension capabilities to ensure accurate classification.

**Table 3. Classification of vaccine stance in tweets.**

| Stance | Description |
|---|---|
| Pro-vaccine | The tweet explicitly supports vaccination, emphasizing its effectiveness and safety, or encourages others to get vaccinated. For example, mentioning that vaccines are essential tools for preventing diseases. |
| Anti-vaccine | The tweet explicitly opposes vaccination, possibly questioning the safety and efficacy of vaccines, or opposing mandatory vaccination policies. This includes highlighting vaccine side effects or framing vaccination as an infringement on personal freedom. |
| Neutral | The tweet does not express a clear stance but discusses vaccine-related policies or scientific information without making a judgment. |
| No stance mentioned | The tweet does not mention any vaccine stance or is unrelated to vaccines. |

## Anti-vaccine theme categorization

To examine whether policy termination influenced the structure of anti-vaccine discourse, we classified anti-vaccine tweets into eight thematic categories using GPT-4o, following the topic taxonomy established by Poddar et al. [8]. These eight themes were originally derived through an iterative process combining Latent Dirichlet Allocation for initial topic discovery, human expert consolidation of candidate topics with seed words, and Labelled-LDA [47] for refined topic assignment. In our study, we directly applied these predefined themes as classification labels and employed GPT-4o to assign each anti-vaccine tweet to the most relevant category. Table 4 presents the eight anti-vaccine theme categories with definitions and examples.

## Stance consistency analysis

For quote tweets (QT), stance consistency was assessed by comparing the poster's comment with the quoted tweet. Each QT was classified into one of three categories: consistent, inconsistent, or unclear, as shown in Table 5. GPT-4o performed this classification based on context-aware reasoning, analyzing both the quoted tweet and the poster's added comment. Since social media language often includes implicit messaging, sarcasm, and figurative speech, LLM-based analysis may encounter challenges in determining stance consistency. In cases where clear intent could not be inferred from the text, the tweet was labeled as unclear.

## Annotation validation

To assess the reliability of GPT-4o classification, we conducted two validation procedures.

**External benchmark validation.** We tested our stance classification prompt on the CAVES dataset [48], a COVID-19 vaccine stance benchmark developed by the same research team that established the thematic taxonomy we adopted [8]. CAVES contains 1,977 expert-labeled tweets with stance annotations. Our classifier achieved $F1 = 0.883$ for anti-vaccine stance identification (Precision = 0.951, Recall = 0.824), demonstrating strong generalization beyond our study dataset.

**Cross-annotator agreement.** We randomly sampled 500 tweets from our dataset and obtained independent annotations from two additional annotators using identical classification criteria. Inter-annotator agreement measured by Cohen's $\kappa$ ranged from 0.71 to 0.79 for stance classification and 0.72 to 0.86 for theme classification, indicating substantial agreement [49].

**Table 4. Thematic categorization of anti-vaccine tweets.**

| Theme | Description |
|---|---|
| **Deeper conspiracy** | Suggests that vaccines are tools for global control or surveillance, such as claims about population control or microchip implantation. |
| **Health concerns** | Raises concerns about vaccine side effects, health risks, or deaths, questioning vaccine safety. |
| **Against mandatory vaccination** | Argues that vaccine mandates infringe on personal freedom and advocates for individual choice. |
| **Ineffective** | Questions vaccine efficacy, claiming that vaccines do not provide sufficient immune protection. |
| **Rushed** | Claims that vaccines were inadequately tested and lack long-term safety data. |
| **Shedding** | Believes that vaccinated individuals may spread the virus through contact. |
| **Big Pharma** | Argues that vaccine promotion is driven by corporate profits rather than public health. |
| **Political** | Claims that vaccine policies are politically motivated and serve specific political agendas. |

**Table 5. Criteria for stance consistency classification in quote tweets.**

| Stance consistency | Description |
|---|---|
| **Consistent** | The poster's stance aligns with the original tweet. For example, a pro-vaccine poster quoting another pro-vaccine tweet. |
| **Inconsistent** | The poster's stance contradicts the original tweet. For example, an anti-vaccine poster quoting and criticizing a pro-vaccine tweet. |
| **Unclear** | The stance of the poster cannot be clearly inferred from the text, or the poster does not explicitly express a stance. |

## Statistical analysis

This study employs multiple logistic regression models to evaluate the association between Musk's termination of X's COVID misinformation policy and changes in anti-vaccine discourse, focusing on changes in the proportion of anti-vaccine tweets and the distribution of anti-vaccine topics. All models are estimated separately for four types of tweets: total population, original tweets, quoted tweets, and retweets.

We construct two sets of logistic regression models to address distinct research questions. The first set examines whether policy termination was associated with a rise in anti-vaccine tweets. In these models, the dependent variable ($Y$) is a binary indicator, where $Y=1$ if the tweet expresses an anti-vaccine stance and $Y=0$ if it supports vaccination or remains neutral. The second set of models investigates whether policy termination was associated with changes in the thematic distribution of anti-vaccine tweets. Here, the dependent variable ($Y$) is also binary, with $Y=1$ indicating that the tweet pertains to a specific anti-vaccine topic and $Y=0$ meaning the tweet does not address that topic. Since each topic is modeled separately, this results in multiple logistic regression analyses.

The independent variable in all models is policy change ($X_{policy}$), where $X_{policy}=0$ represents the pre-policy termination period (November 16–22, 2022), and $X_{policy}=1$ represents the post-policy termination period (November 24–30, 2022). All models report odds ratios, 95% confidence intervals, and p-values, with statistical significance determined at $p<0.05$.

Thus, each logistic regression follows the general form shown in Eq (1):

$$\log\left(\frac{P(Y=1)}{P(Y=0)}\right) = \beta_0 + \beta_1 X_{policy}$$

(1)

where $Y$ represents the dependent variable, which varies based on the model type as described above. This analysis aims to reveal whether the overall prevalence and thematic structure of anti-vaccine discourse changed following policy termination.

To assess the robustness of our findings, we also conducted sensitivity analyses excluding highly retweeted content (≥ 50 retweets) to examine whether results were driven by a small number of widely shared posts.

## Results

### Exploratory analysis of identified vaccine stances and topics

The final annotated dataset comprised 13,458 tweets: 5,896 from the pre-policy period (November 16–22) and 7,562 from the post-policy period (November 24–30). By tweet type, the dataset included 2,444 original tweets (18.2%), 10,001 retweets (74.3%), and 1,013 quote tweets (7.5%). Regarding vaccine stance, 6,674 tweets (49.6%) were classified as anti-vaccine, 3,949 (29.3%) as neutral, 2,442 (18.1%) as pro-vaccine, and 393 (2.9%) had no vaccine stance mentioned. This distribution yielded a near-balanced classification between anti-vaccine (49.6%) and non-anti-vaccine content (50.4%).

This subsection presents an exploratory analysis of vaccine stance tweets annotated by GPT-4o, examining trends before and after X terminated its COVID-19 misinformation policy. Fig 1 illustrates the temporal trends of different vaccine stances, highlighting a pronounced increase in tweet volume immediately following the policy change (Nov 23). Notably, this increase was driven largely by anti-vaccine tweets, whereas pro-vaccine and neutral tweets exhibited relatively smaller changes. Fig 2 further decomposes these trends across original tweets (TW), retweets (RT), and quote tweets (QT), showing a particularly marked rise in retweet activity. This indicates that the policy shift mainly amplified existing vaccine discourse rather than prompting new original content creation. Fig 3 reinforces this pattern by quantitatively comparing tweet types, clearly revealing that retweeting behavior was the primary mechanism amplifying vaccine-related narratives post-policy termination.

To further contextualize these visual patterns, Tables 6 and 7 provide examples and thematic distributions of non-anti-vaccine and anti-vaccine tweets before and after the policy shift. Among non-anti-vaccine tweets, neutral content increased proportionally, whereas explicitly pro-vaccine tweets showed a modest decrease. Anti-vaccine tweets exhibited notable thematic shifts, with health concerns, vaccine inefficacy, and conspiracy narratives becoming more prominent after the policy termination. Fig 4 shows the overall distribution of anti-vaccine themes, while Fig 5 illustrates how the thematic composition evolved over time before and after the policy change.

### Does policy termination lead to an increase in anti-vaccine tweets?

To statistically evaluate whether X's termination of the COVID-19 misinformation policy was significantly associated with the prevalence of anti-vaccine tweets, we conducted regression analyses (Fig 6). The results indicate that the policy termination was significantly associated with an increase in anti-vaccine tweets within the total population of analyzed tweets (odds ratio [OR] = 1.60, 95% CI: 1.50–1.72, $p < 0.001$). Specifically, retweets exhibited the most substantial increase in anti-vaccine sentiment following policy termination (OR = 1.75, 95% CI: 1.62–1.90, $p < 0.001$), highlighting retweets as the primary mechanism amplifying existing anti-vaccine content. Original tweets also showed a statistically significant, albeit more moderate, increase in anti-vaccine stance (OR = 1.41, 95% CI: 1.19–1.67, $p < 0.001$). Conversely, quoted tweets did

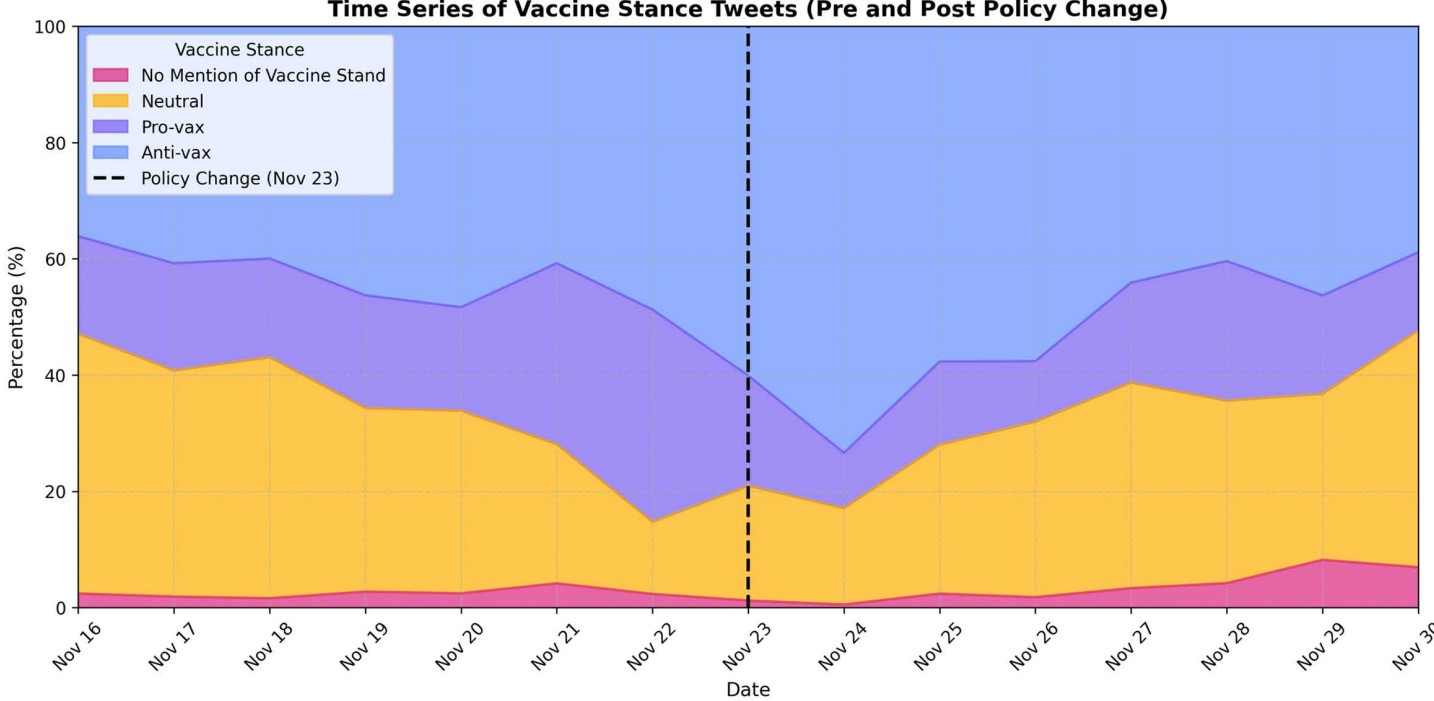

**Fig 1. Temporal distribution of vaccine stance tweets before and after policy change.** Each color represents a distinct stance category: anti-vaccine, neutral, pro-vaccine, and no mention of vaccine stance. The dashed line denotes the policy change date (Nov 23). The y-axis shows the percentage of each stance category, highlighting the relative shift in composition following the policy update.

not demonstrate a statistically significant change (OR = 0.98, 95% CI: 0.77–1.26, $p = 0.893$), indicating no meaningful shift in anti-vaccine stance among quote tweets following the policy termination.

To address concerns about highly retweeted content disproportionately influencing results, we conducted a sensitivity analysis by excluding tweets that were retweeted 50 or more times. This filtering removed 30 original tweets and their 5,407 associated retweets (40.2% of the data). Table 8 presents representative examples of highly retweeted anti-vaccine content.

The sensitivity analysis results reveal an important nuance: while the overall effect of policy termination remained significant, the retweet-specific effect was substantially attenuated and no longer statistically significant. Notably, the effect for original tweets increased, suggesting that the policy change was associated with the creation of new anti-vaccine content rather than merely amplifying existing narratives. This finding underscores the importance of distinguishing between content amplification and content generation when assessing policy impacts on online discourse (Fig 7).

### Does the policy termination significantly shift the prevalence of anti-vaccine themes?

To assess whether X's termination of its COVID-19 misinformation policy was significantly associated with the thematic prevalence of anti-vaccine tweets, we conducted regression analyses focusing on four predominant anti-vaccine themes: health concerns, deeper conspiracy, opposition to mandatory vaccination, and vaccine ineffectiveness. Fig 8 presents the complete statistical results.

Our results reveal distinct thematic shifts following policy termination. Tweets emphasizing health concerns significantly increased overall, particularly driven by retweets and quoted tweets, while original tweets showed no statistically

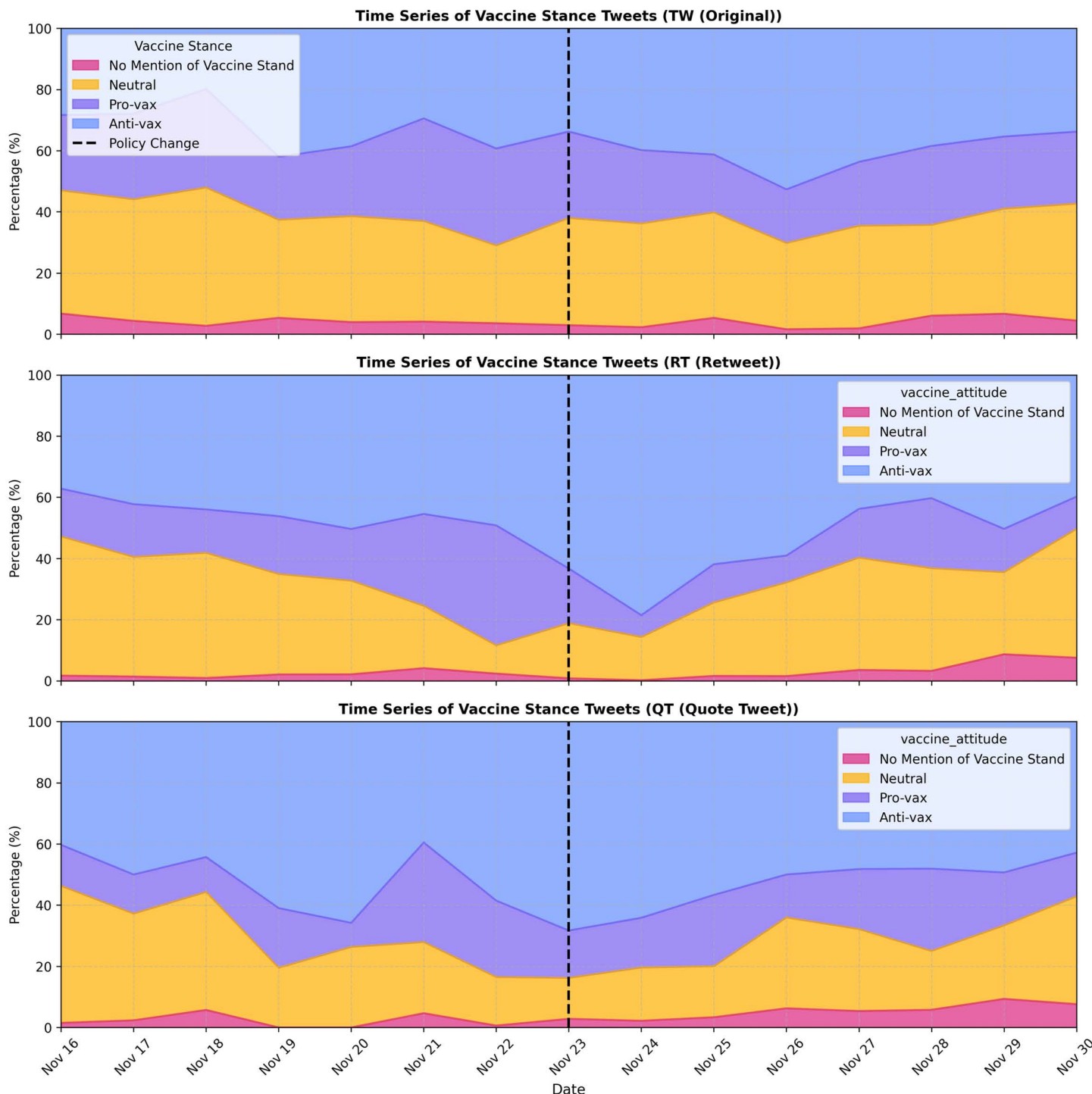

**Fig 2. Temporal trends in vaccine stance tweets by tweet type.** These three subplots illustrate vaccine stance trends across original tweets (TW), retweets (RT), and quote tweets (QT). The dashed line represents the policy change date. The y-axis shows the percentage of each stance category within each tweet type.

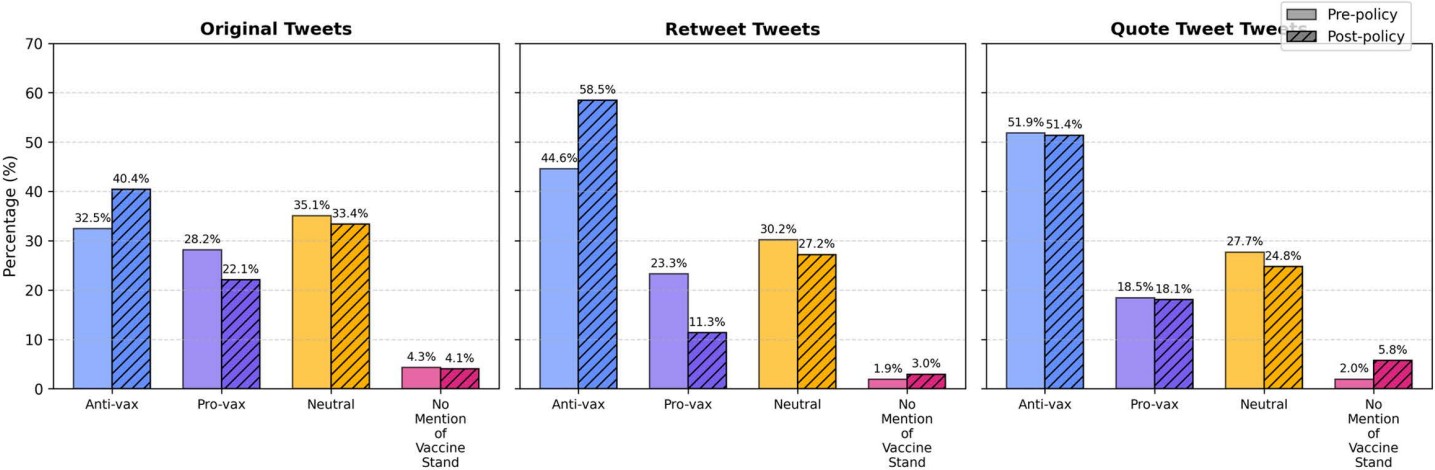

**Fig 3. Percentage distribution of vaccine stances across tweet types before and after policy change.** Solid bars represent pre-policy data, while hatched bars indicate post-policy data. Percentages are shown above each bar, facilitating direct comparison of compositional shifts across tweet types.

**Table 6. Examples and distribution of non-anti-vaccine tweets before and after policy termination.**

| Pre-policy (before Nov 23) | Post-policy (after Nov 23) |
|---|---|
| **Pro-vaccine:** Explicitly supports vaccination. | |
| **[41.7%]** @BBCNews If COVID-19 is anywhere, it is potentially everywhere. COVID-19 vaccines will save many lives. Mask wearing is also a good idea. | **[30.4%]** Woohoo! I finally was about to get my COVID-19 booster and flu shot. I had to wait three months because I caught COVID in August. Please mask up and get your vaccine if you're able to. Stay safe! |
| **Neutral:** Mentions vaccines without taking a stance. | |
| **[54.1%]** Injunction sought over use of COVID-19 vaccines on children aged 5–11 | **[62.2%]** Serious adverse events of special interest following mRNA COVID-19 vaccination in randomized trials in adults |
| **No mention of vaccine stance:** May discuss unrelated topics. | |
| **[4.2%]** Abbott: Texas schools forbidden to require COVID-19 vaccines for students | **[7.4%]** Kim Jong un. North Korean president. Person wey order the 1st covid19 case in his country to be shot dead. |

significant change. This indicates that the policy shift notably amplified user engagement in disseminating existing health-risk narratives rather than generating new content on this theme.

In contrast, tweets associated with deeper conspiracy narratives and opposition to mandatory vaccination both exhibited significant overall decreases, primarily driven by reduced retweeting activity. Original tweets and quoted tweets for these themes showed no significant change, suggesting that while these narratives were less frequently amplified, their original creation remained stable.

The vaccine ineffectiveness theme presented a more nuanced pattern. While overall prevalence decreased, original tweets on this theme showed a statistically significant increase. This suggests that users were more likely to generate new content questioning vaccine effectiveness post-policy termination, even as the overall retweeting of such content declined.

**Table 7. Anti-vaccine tweet topics, distribution, and examples before and after policy termination.**

| Pre-policy (before Nov 23) | Post-policy (after Nov 23) |
|---|---|
| **Topic 1: Health concerns** Health and safety issues, including concerns about vaccine ingredients, side effects, and deaths. | |
| [32.1%] #wednesdaythought The Brits know it, how long will it take Americans to understand that the vaccine is killing more people than #COVID19? | [47.0%] Moderna COVID-19 vaccine is an mRNA type, which contains genetically engineered cells that can permanently alter human DNA. |
| **Topic 2: Against mandatory** Opposition to mandatory vaccination, with discussion of freedom and choice. | |
| [29.2%] The incoming GOP House majority can defund every executive order. I would start with the COVID-19 vaccine mandates that are unconstitutional and illegal. | [21.0%] SAY NO TO COVID-19 VACCINES GET YOUR DIGITAL PASSPORT and COVID-19 vaccine cards, BOOSTERS, EXEMPTION LETTER without taking the vaccine Click the link below |
| **Topic 3: Big Pharma** Distrust in pharmaceutical companies, including accusations of deception and profit motives. | |
| [2.3%] This is just absolutely incredible. I guess the vax companies are gonna clook into it now. | [8.6%] So the vaccinated account for a majority of COVID-19 deaths. So, is it ok to say now that the vaccine was a money grab for the pharmaceutical companies? FYI we still unvaccinated and proud. |
| **Topic 4: Political** The vaccine is political, with governments pushing their own agendas. | |
| [7.4%] When it comes to the COVID-19 vaccines, Trump is in lala land. He was fooled, and he still either hasn't realized it, or can't admit it. Either way, he's moving further away from presidential material. | [6.3%] Yet Biden called Trudeau and demanded he crush the Trucker's protest against COVID-19 vaccine mandates. Then Trudeau did just that, using unconstitutional emergency powers. |
| **Topic 5: Ineffective** Skepticism about vaccine effectiveness. | |
| [11.0%] Do people actually want a universal vaccine? Seems like a hard sell after the failed COVID-19 injections. Waste of time and money if you ask me. | [8.0%] Over time, COVID-19 shots increase rather than decrease the risk of contracting and spreading the virus. |
| **Topic 6: Rushed** Concerns that vaccine development was rushed and inadequately tested. | |
| [4.7%] They told us that the long-term side effects of COVID-19 were unknown. Then in the same breath said we should take a vaccine that was rushed to market and not worry about the side effects. #vaccineinjuries #VaccineDeaths | [1.5%] @MarcLobliner Vaccines take years, sometimes decades, to develop, yet they came up with one in the same year as the COVID-19 outbreak. Yeah, it was nothing more than an experiment. No thanks. |
| **Topic 7: Shedding** Concerns that vaccinated individuals transmit the virus. | |
| [0.0%] Vax Shedding Acknowledged by Pfizer in 2020 #COVID19 #vaccine | [0.3%] More people have died after vaccines came into play than under Trump!! The shedding of COVID-19 vaccines has infected more people than we know. |
| **Topic 8: Deeper conspiracy** The disease is a hoax, tracking chips, and other conspiracy theories. | |
| [13.3%] Knock knock. Who's there? DARPA. DARPA who? DARPA whose mandate is to make vast weapon systems of the future and who made the mRNA injections forced on us. Pass it on. | [7.3%] Who are these? The ones responsible for the crimes against humanity being committed right now with the continued pushing of the gene therapy called COVID-19 vaccine. |

## Impact of policy termination on stance consistency in quote tweets

We further analyzed whether X's policy termination impacted the consistency of vaccine stances expressed by users in quote tweets relative to the original quoted content (Fig 9). Overall, anti-vaccine quote tweets exhibited the highest stance consistency (255 tweets), while neutral tweets were most frequently classified as unclear (152 tweets), indicating challenges in accurately interpreting neutral user attitudes from their commentary.

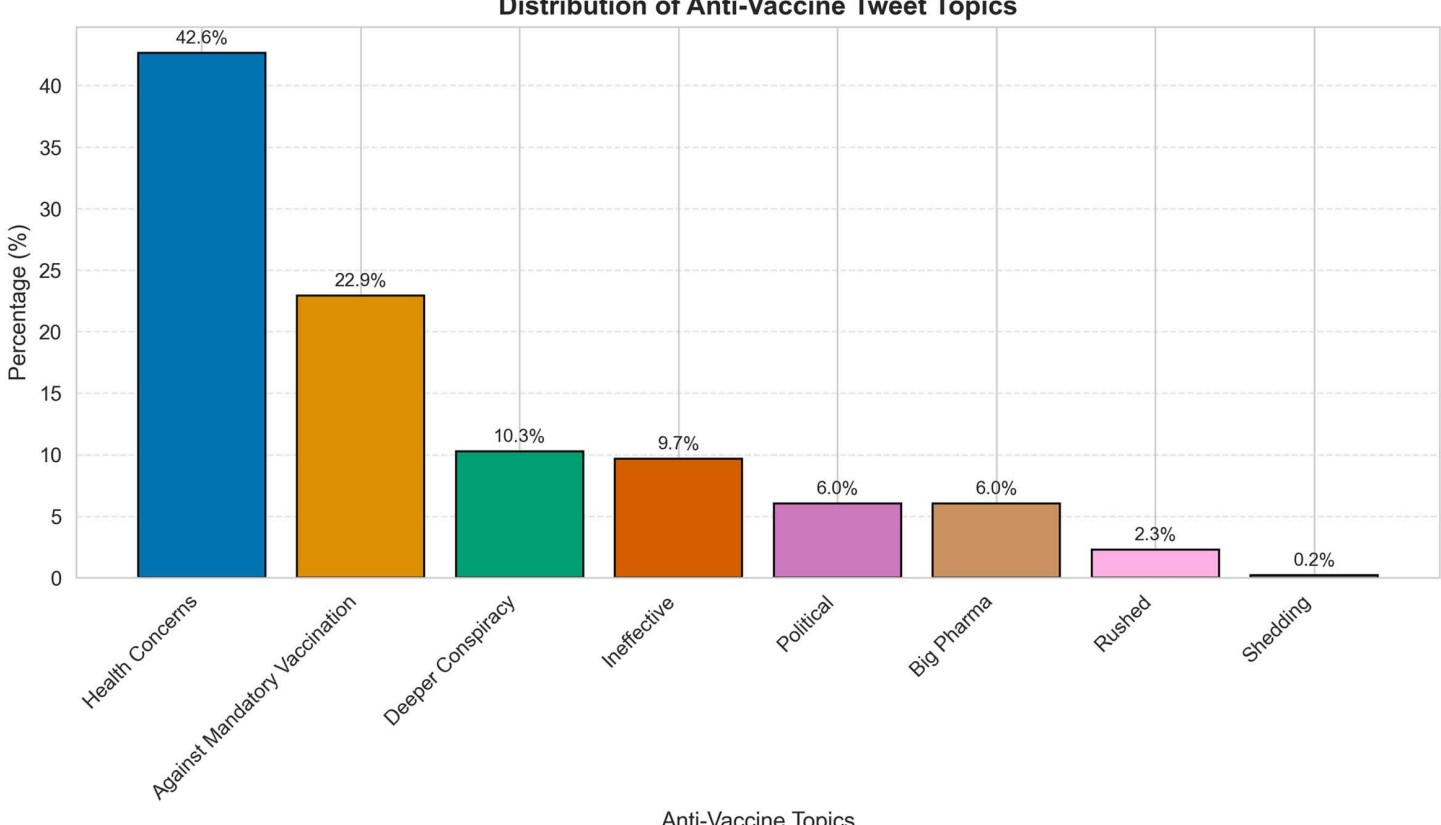

**Fig 4. Percentage distribution of anti-vaccine themes.** The most prevalent theme is health concerns, followed by against mandatory vaccination and deeper conspiracy. Less common topics include shedding and Big Pharma. Percentages are shown above each bar, illustrating the relative prevalence of vaccine hesitancy narratives.

Comparing tweet distributions before and after policy termination reveals notable shifts. Before the policy change, anti-vaccine quote tweets showed nearly equal proportions of consistent (109 tweets) and inconsistent stances (100 tweets), with fewer unclear cases (57 tweets). After policy termination, there was a substantial increase in consistent anti-vaccine stances (146 tweets), accompanied by a marked decrease in inconsistent stances (41 tweets). The unclear cases among anti-vaccine tweets also increased (72 tweets), though less markedly. This pattern suggests a growing tendency among users to reinforce existing anti-vaccine messages after content moderation ceased.

Neutral quote tweets maintained a predominantly unclear stance both before (80 tweets) and after (72 tweets) policy termination, suggesting persistent ambiguity in neutral vaccine discussions. Pro-vaccine quote tweets exhibited relatively stable distributions over time, consistently showing a clear alignment with quoted content.

## Discussion

### Principal findings

This study examined the changes following X's termination of its COVID-19 vaccine misinformation policy on vaccine-related discourse, addressing three research questions. For anti-vaccine tweet volume (RQ1), our findings indicate that the proportion of anti-vaccine tweets significantly increased after the policy change, particularly in retweets and quote tweets. Meanwhile, the proportion of pro-vaccine tweets declined, whereas neutral tweets became more prevalent. This

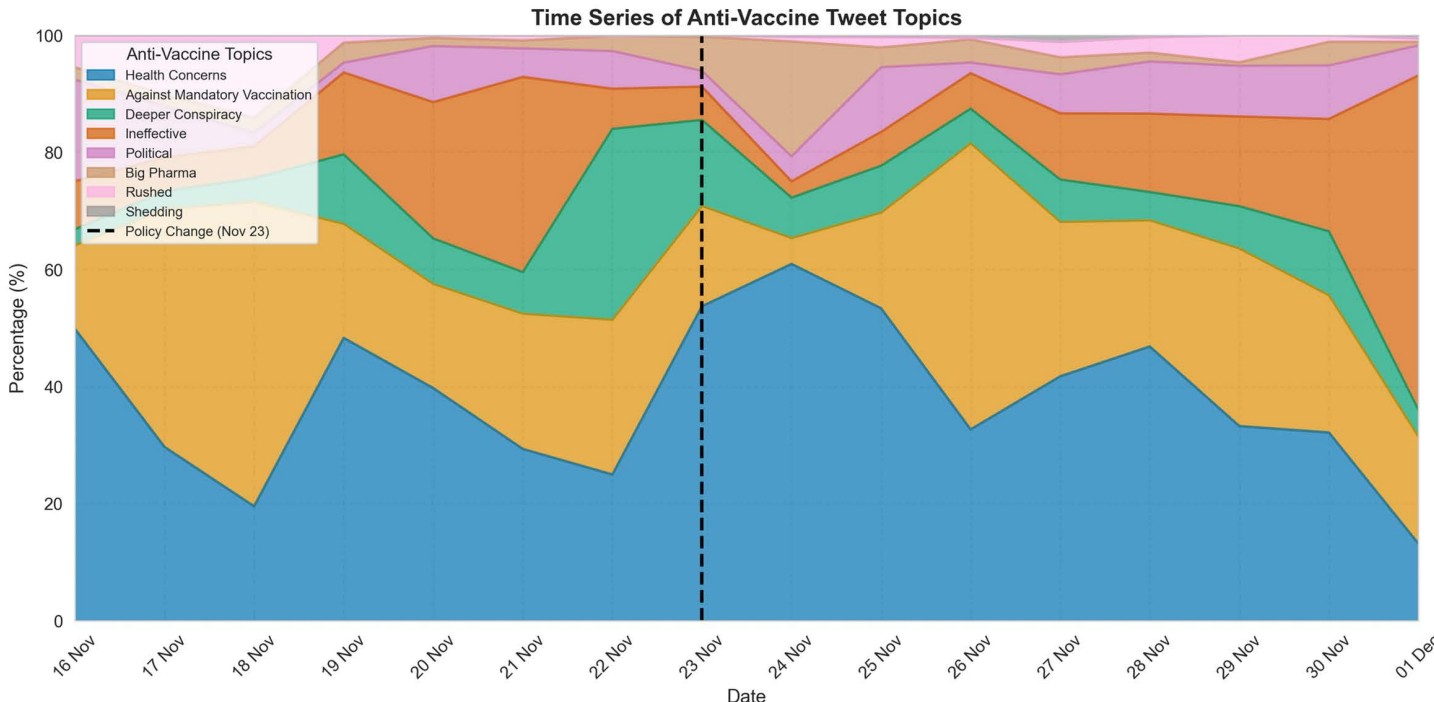

**Fig 5. Time series of anti-vaccine tweet topics from Nov 16 to Nov 30.** Each color represents a different theme, and the dashed line marks the policy change date (Nov 23). The y-axis shows the percentage composition of topics, revealing how the relative prominence of health concerns, government conspiracies, and vaccine inefficacy narratives shifted following the policy change.

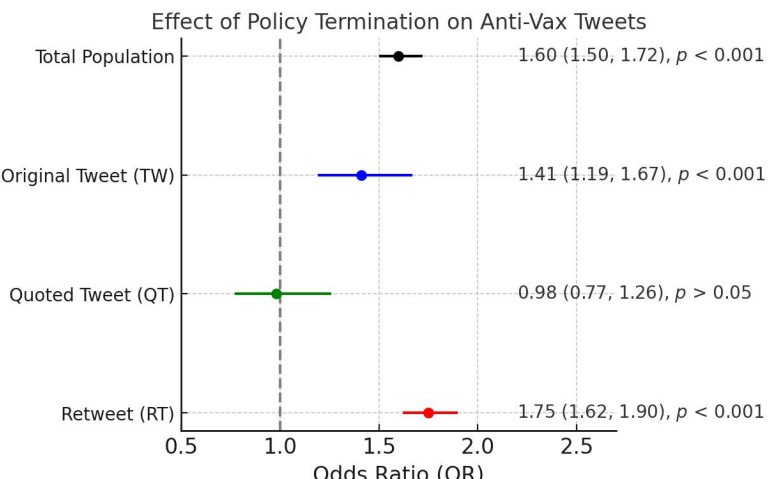

**Fig 6. Association between policy termination and anti-vaccine tweets.** Odds ratios with 95% confidence intervals are presented for different tweet types. The overall population exhibits a significant increase in anti-vaccine tweets, while quoted tweets show no statistically significant change.

**Table 8. Examples of highly retweeted anti-vaccine tweets during the study period.**

| Date | Retweets | Content summary |
|------|----------|-----------------|
| Nov 23 | 605 | Promoted "Died Suddenly" documentary, appealing directly to Elon Musk not to censor it |
| Nov 22 | 340 | Cited pseudo-scientific claims linking Massa-chusetts death certificates to vaccine deaths |
| Nov 25 | 243 | Attacked Dr. Anthony Fauci, framing unvacci-nated individuals as victims rather than threats |

suggests that the policy adjustment was associated with changes in the volume of anti-vaccine content but also altered the overall information landscape, where pro-vaccine discourse became relatively less visible. Notably, sensitivity analyses excluding highly retweeted content revealed that the policy change was associated with increased creation of new anti-vaccine content, rather than merely amplifying existing narratives through retweets.

Importantly, our primary analysis intentionally retained retweets because they constitute the information environment to which users are actually exposed. From a public health perspective, it is the prevalence of anti-vaccine content in users' feeds, regardless of whether it originates from new posts or amplified existing ones, that shapes exposure and potentially influences attitudes [3]. Our sensitivity analysis complements this by distinguishing between content amplification and content generation, addressing whether the observed increase reflects viral spread of a few posts or broader discourse changes. Together, these analyses provide a more complete picture: the main analysis captures changes in the information landscape users encounter, while the sensitivity analysis confirms that the policy change was associated with both amplification of existing content and creation of new anti-vaccine discourse.

For thematic shifts (RQ2), the results show that the thematic distribution of anti-vaccine tweets changed after the policy termination. Specifically, tweets expressing concerns about vaccine safety and health risks became more prominent, while changes in themes related to distrust in pharmaceutical companies and opposition to vaccine mandates were relatively minor. Additionally, some conspiracy-based narratives, such as those linking vaccines to government control or microchip implantation, declined. These findings suggest that after the policy termination, anti-vaccine discourse increasingly centered on vaccine safety concerns rather than adopting more extreme conspiracy narratives.

For stance consistency in quote tweets (RQ3), our analysis found that after the policy termination, anti-vaccine users were more likely to quote tweets aligned with their own stance and less likely to engage with opposing views. This suggests that the policy change not only increased the volume of anti-vaccine tweets but also reinforced ideological polarization, leading to decreased cross-stance interactions in vaccine discourse.

## Comparison with prior literature

Our findings align with and extend prior research on the dissemination of vaccine discourse on social media. Previous studies have shown that anti-vaccine narratives often rely on emotionally charged and polarized rhetoric, making them more engaging and widely shared than pro-vaccine content [19,20]. Our study further confirms this trend, showing that anti-vaccine tweets became even more pronounced after the policy change, particularly through retweets [28,29]. This amplification pattern is consistent with Mønsted and Lehmann's [27] characterization of polarization in online vaccine discourse.

Regarding the impact of content moderation policies, prior research has primarily focused on their implementation and effectiveness in reducing the spread of misinformation. Studies have shown that interventions such as content labeling and fact-checking can reduce engagement with misleading narratives [9,31]. However, fewer studies have examined the short-term effects of policy removal [41]. Our study fills this gap by documenting that the termination of X's misinformation policy did not lead to a uniform increase in anti-vaccine discourse but rather a

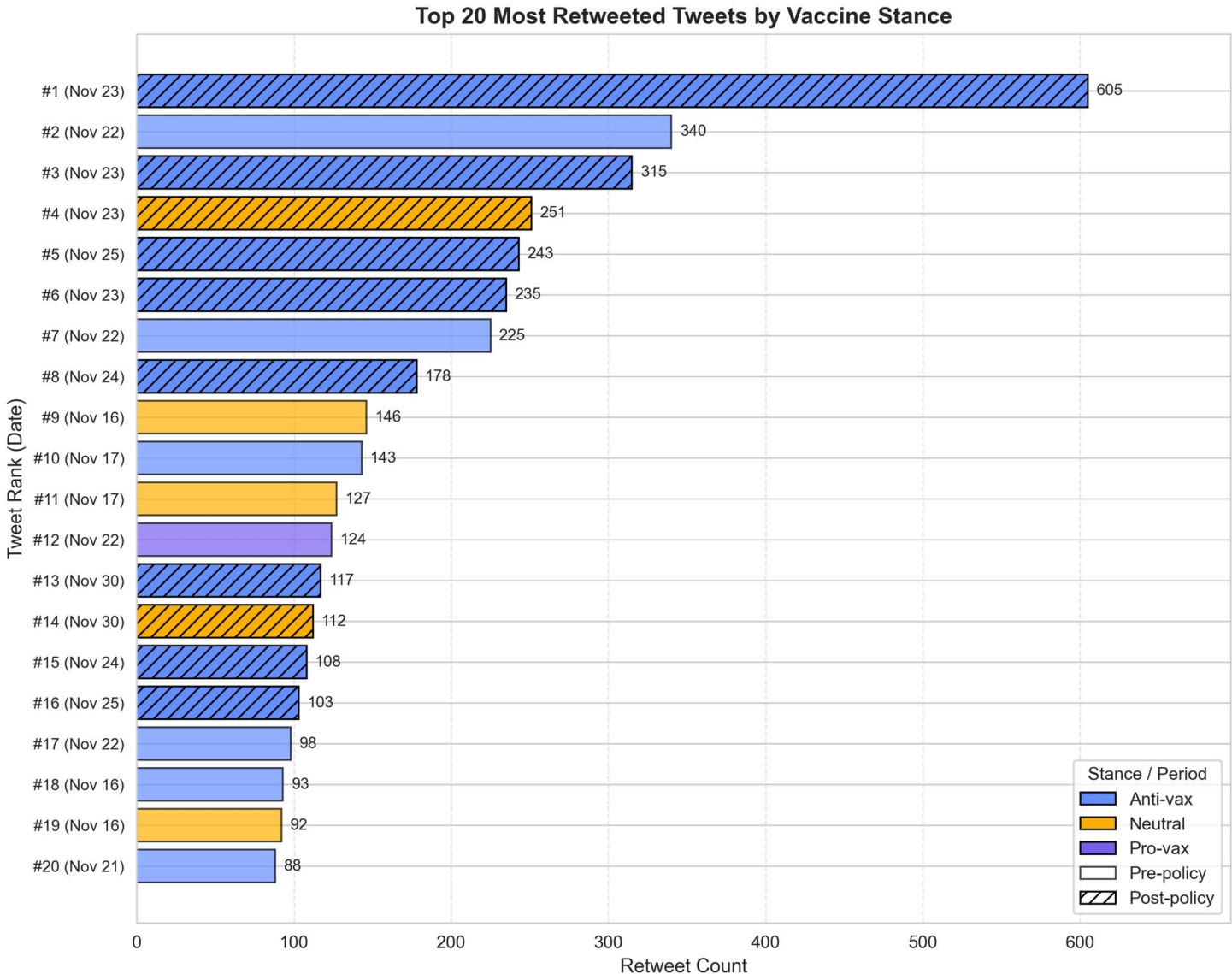

**Fig 7. Top 20 most retweeted tweets during the study period, colored by vaccine stance.** Hatched bars indicate tweets from the post-policy period (Nov 24–30). Despite having shorter observation windows for accumulating retweets (0–6 days versus 8–14 days for pre-policy tweets), post-policy tweets comprised 50% of the top 20 most retweeted tweets, suggesting accelerated spread of anti-vaccine content following policy termination.

selective amplification of specific themes, particularly vaccine safety concerns. This aligns with the findings of Allen et al. [24], who suggested that vaccine-skeptical narratives, rather than outright false claims, play a stronger role in influencing vaccine hesitancy.

Our findings on stance consistency complement [42], who showed that Facebook's moderation drove anti-vaccine content toward politicization. We observed a different pattern: increased ideological alignment in quote tweets, suggesting that policy removal was associated with changes in user interaction patterns rather than just content volume.

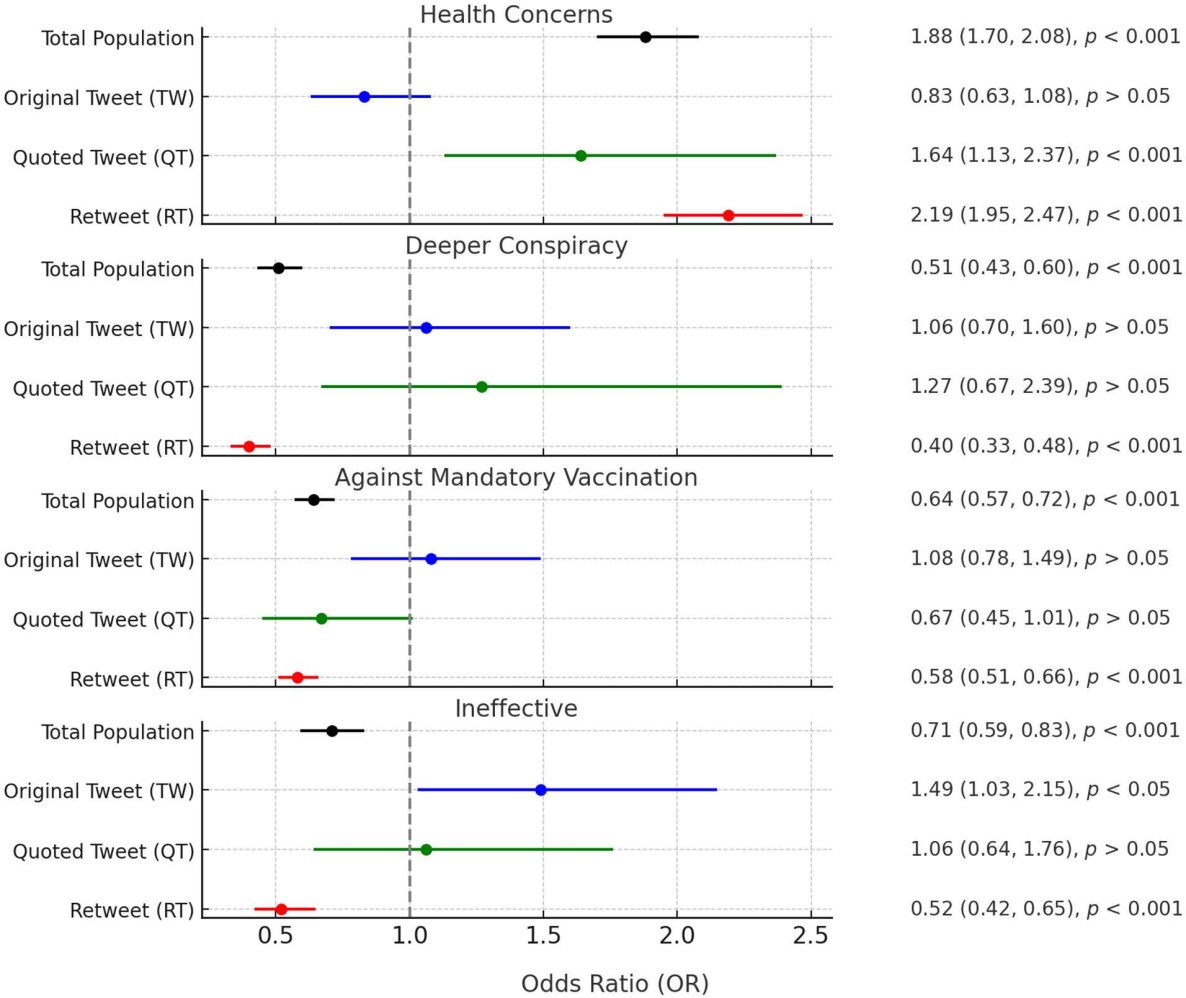

**Fig 8. Odds ratios for anti-vaccine themes across tweet types before and after policy change.** Each panel corresponds to a distinct anti-vaccine narrative, showing whether policy change was associated with its prevalence. Statistically significant increases and decreases are observed for various tweet types.

## Methodological contributions

A key contribution of this study is demonstrating the feasibility of using large language models for large-scale social media content analysis. We employed GPT-4o with Chain-of-Thought prompting to perform three classification tasks, stance detection, thematic categorization, and stance consistency assessment, on over 13,000 tweets. This approach offers several advantages over traditional methods.

First, compared to manual annotation, LLM-based classification enables analysis at scale while maintaining consistency. Our validation on the CAVES benchmark [48] achieved F1 = 0.883 for anti-vaccine stance detection, with cross-annotator agreement (Cohen's $\kappa$ = 0.71–0.86) comparable to human inter-rater reliability. Second, unlike keyword-based or traditional machine learning approaches that require extensive labeled training data, our prompt-based method can be

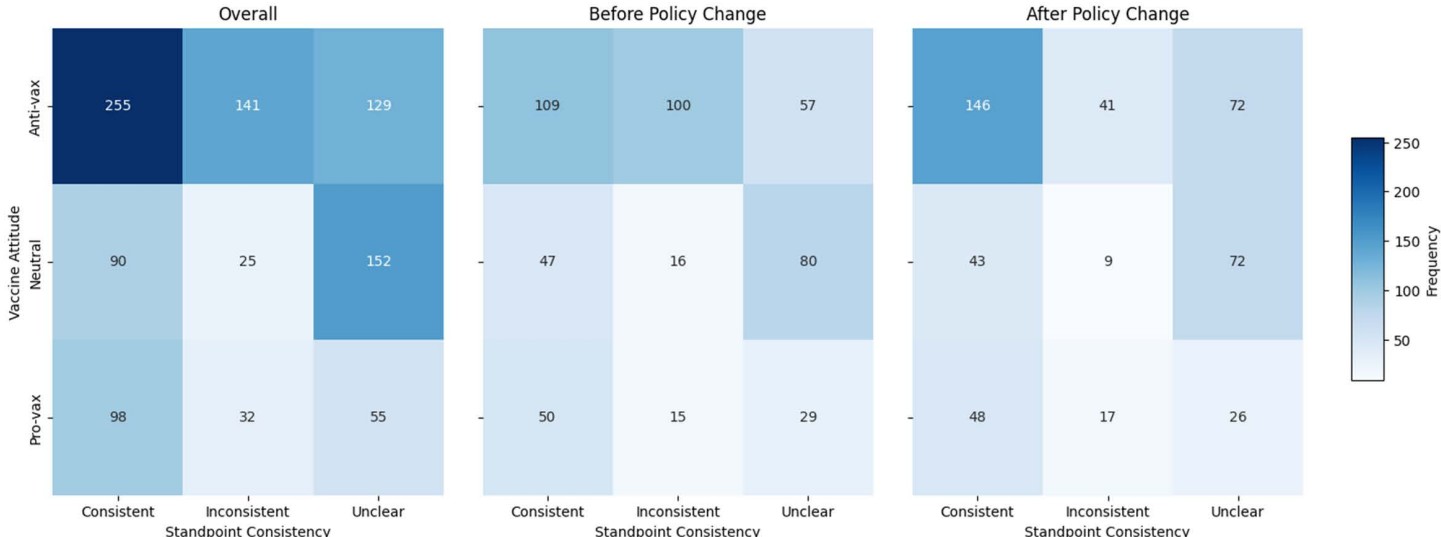

**Fig 9. Distribution of vaccine attitudes against standpoint consistency before and after policy change.** The leftmost heatmap represents the overall distribution, while the middle and right heatmaps correspond to the pre-policy and post-policy periods, respectively. The color intensity reflects the frequency of each category, with darker shades indicating higher tweet counts. Notably, anti-vaccine quote tweets show increased alignment with the quoted content after the policy change.

rapidly adapted to new classification tasks. Third, the Chain-of-Thought prompting provides interpretable reasoning for each classification decision, enhancing transparency.

These methodological advances complement recent work on automated misinformation detection [50] and suggest that validated LLM classifiers can serve as practical tools for researchers studying online discourse dynamics, particularly when events unfold rapidly and timely analysis is needed.

## Policy implications

Our findings have implications for platform governance. Within this short pre-post window, the termination of X's COVID-19 misinformation policy was associated with greater visibility of anti-vaccine content, especially through retweets. The observed thematic shifts also indicate that this policy change was associated with different short-term patterns across narrative types. Increased stance consistency in quote tweets further suggests greater within-stance alignment in vaccine-related discussion.

These findings highlight the complexity of content moderation decisions. In this case, terminating X's COVID-19 misinformation policy was associated with short-term shifts in the visibility and composition of vaccine-related discourse. At the same time, our observational design does not permit causal inference. The broader question of how platforms should balance free expression and public health remains open.

## Limitations and future directions

Despite the insights provided by this study, several limitations should be acknowledged. First, our analysis is based on a short time window covering only seven days before and after the policy termination, which does not allow for assessment of long-term effects. Additionally, pre-policy tweets had more time to accumulate retweets than post-policy tweets, yet post-policy content still comprised 50% of the top 20 most retweeted tweets (Fig 7), suggesting our estimates may be conservative. Future research should extend the observation period to determine whether the observed changes persist over time.

Second, our pre-post observational design precludes causal inference, as the period surrounding the policy termination was characterized by multiple concurrent events. Our analysis captures changes associated with the announcement of policy termination rather than changes in enforcement practices. Relatedly, our regression models use a single binary predictor (pre/post policy) without additional control variables. This parsimonious specification reflects our short 14-day observation window, during which user composition and platform characteristics are unlikely to change systematically. Stratified analyses by tweet type partially address confounding, but potential confounders such as concurrent news events or algorithm changes cannot be adequately captured through tweet-level covariates. Furthermore, this study focuses on tweet content without analyzing user-level behavior, such as which users were more likely to post or amplify anti-vaccine content, or the role of key opinion leaders. Prior research has shown that different user categories, such as government accounts, media outlets, and ordinary users, play distinct roles in epidemic-related information propagation, with media dominating reach while general users propagate information faster [51]. Future studies could integrate user-level analysis to better understand differential responses to policy changes.

Finally, our study does not explore the social network structures underlying the spread of anti-vaccine discourse. Future research could employ social network analysis to examine how policy termination influenced information diffusion pathways and community dynamics.

## Conclusion

This study examined changes in vaccine-related discourse following X's termination of its misinformation policy, focusing on the prevalence, thematic composition, and stance consistency of anti-vaccine tweets. Our findings indicate that the policy change was associated with a significant increase in anti-vaccine content, particularly through retweets, which served as the primary mechanism for amplification. Sensitivity analyses further revealed that the policy change was associated with both the amplification of existing content and the creation of new anti-vaccine discourse. Thematic analysis indicated that while concerns over vaccine safety intensified, deeper conspiracy narratives and opposition to vaccine mandates declined. Additionally, stance consistency in quote tweets strengthened post-policy termination, suggesting a reinforcement of ideological alignment within anti-vaccine discourse.

These findings contribute to the ongoing debate on content moderation by suggesting that the removal of misinformation policies may be associated not only with an increase in the quantity of controversial discourse but also with changes in thematic structure and user engagement patterns. The results highlight the role of platform governance in shaping public health narratives. However, our pre-post observational design and the presence of concurrent events during the study period preclude definitive causal conclusions. Future research should extend the observation period to examine the long-term effects of policy changes and investigate the role of key influencers and network dynamics in shaping discourse evolution.

## Author contributions

**Data curation:** Yufei Li, Tianhao Chen.

**Methodology:** Tianhao Chen, Wei Ke.

**Project administration:** Yufei Li, Patrick Pang.

**Resources:** Patrick Pang, Shanton Chang, Nancy Baxter.

**Supervision:** Wei Ke, Patrick Pang, Dana McKay, Shanton Chang.

**Validation:** Yufei Li.

**Visualization:** Yufei Li.

**Writing – original draft:** Yufei Li, Tianhao Chen.

**Writing – review & editing:** Yufei Li, Tianhao Chen, Yanjie Zhao, Dana McKay, Nancy Baxter.

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
