## [Decision Letter · Decision Letter 0]

13 Oct 2025

Dear Dr. Pang,

Thank you for submitting your manuscript to PLOS ONE. After careful consideration, we feel that it has merit but does not fully meet PLOS ONE’s publication criteria as it currently stands. Therefore, we invite you to submit a revised version of the manuscript that addresses the points raised during the review process.

For acceptance, please validate the LLM labels with a small human double-coding and report precision/recall/F1; show counts and proportions with odds ratios and 95% CIs; add simple robustness checks for outliers, retweets/duplicates, and bots; clarify dataset timing, filters, and category scheme; share data/code per PLOS policy; soften causal language; update figures to include proportions with accessible colors; fix reference formatting and typos.

Nice to have: tighten the research gap and early definitions, and expand the discussion to acknowledge alternatives and limits.

Please include a short point-by-point response.

If applicable, we recommend that you deposit your laboratory protocols in protocols.io to enhance the reproducibility of your results. Protocols.io assigns your protocol its own identifier (DOI) so that it can be cited independently in the future. For instructions see: https://journals.plos.org/plosone/s/submission-guidelines#loc-laboratory-protocols. Additionally, PLOS ONE offers an option for publishing peer-reviewed Lab Protocol articles, which describe protocols hosted on protocols.io. Read more information on sharing protocols at. Additionally, PLOS ONE offers an option for publishing peer-reviewed Lab Protocol articles, which describe protocols hosted on protocols.io. Read more information on sharing protocols at. Additionally, PLOS ONE offers an option for publishing peer-reviewed Lab Protocol articles, which describe protocols hosted on protocols.io. Read more information on sharing protocols at. Additionally, PLOS ONE offers an option for publishing peer-reviewed Lab Protocol articles, which describe protocols hosted on protocols.io. Read more information on sharing protocols at https://plos.org/protocols?utm_medium=editorial-email&utm_source=authorletters&utm_campaign=protocols....

We look forward to receiving your revised manuscript.

Kind regards,

Pierluigi Vellucci

Academic Editor

PLOS ONE

Journal Requirements:

3. Please ensure that you refer to Figures 4 and 5 in your text as, if accepted, production will need this reference to link the reader to the figures.

4. We note you have included a table to which you do not refer in the text of your manuscript. Please ensure that you refer to Tables 1 and 4 in your text; if accepted, production will need this reference to link the reader to the Tables.

Reviewers' comments:

Reviewer's Responses to Questions

**Comments to the Author**

1. Is the manuscript technically sound, and do the data support the conclusions?

Reviewer #1: Partly

Reviewer #2: Partly

Reviewer #3: Partly

2. Has the statistical analysis been performed appropriately and rigorously?

Reviewer #1: No

Reviewer #2: Yes

Reviewer #3: Yes

3. Have the authors made all data underlying the findings in their manuscript fully available?

Reviewer #1: No

Reviewer #2: Yes

Reviewer #3: Yes

4. Is the manuscript presented in an intelligible fashion and written in standard English?

Reviewer #1: Yes

Reviewer #2: Yes

Reviewer #3: Yes

Reviewer #1: The paper studies the impact of X's (formerly Twitter) abrupt termination

of its COVID-19 misinformation policy to investigate its effects on anti-vaccine discourse.

It applies pre/post design over a defined period (Nov 16–30, 2022),

using stratified sampling, keyword filtering, and classification via GPT-4o.

The use of GPT-4o for tweet classification and thematic labeling should be better justified.

In section 3.2, it is simply stated that "To verify annotation quality, we randomly sampled 200

labeled tweets for manual review by the research team and found that the LLM performed very well."

This point must be jusified in a more rigorous (statistical) manner.

On top of that, the manuscript does not explicitly state the total number of tweets analyzed

after filtering and annotation; if the classes are balanced and so on...

The paper should be made more self-contained to make it easier to read, especially when dealing

with importants points, eg :

- page 3: "assigned the identified antivaccine tweets into eight thematic categories [8]"

- page 6: "we applied an additional keyword filtering step inspired by [42]"

- page 7: "into eight thematic categories using GPT-4o, following the framework in [8]"

According to section 3.1, the COVID19 dataset is taken from ref. [41] that was published

in 2021. But the dataset of your study goes over the period Nov 16–30, 2022.

How do you explain this mismatch ?

All the figures show raw data (Tweet count). It would be more informative to have it in

percentage. Indeed, if the number of anti-vax Tweets increases together with the number of

pro-vax, what really matters, is the relative proportion of the two.

There are many bugs in the references that should be corrected:

[4] and [28] : the name of the journal does not appear

[10], [13], [14] : question marks appearing in the references

This shows that the bibliographic research was botched. Furthermore, there are numerous

X's studies about the COVID-19 into Plos One, and no one is quoted.

The major flaw of the study, as explained in section 5.4, is the short time Windows, covering

only seven days before and after the policy termination. There is a dissonance with the conclusions

of section 5.3 ("This study provides important insights into content moderation policies on social media

platforms and their broader implications for public health communication"; "X’s misinformation policy may

have played a crucial role in mitigating the spread of anti-vaccine discourse"; "anti-vaccine narratives

gained greater visibility") that should be re-written to take into account the small period of the study.

In other word, the conclusion should be way more modest, in order to be in phase with the study.

Reviewer #2: From my perspective, the title and abstract are both fine, clear, relevant, and well-executed; great work by the authors The Introduction is well-written and clearly structured; however, I suggest it could be improved by explicitly stating the research gap, defining key constructs like stance consistency and amplification earlier, and briefly framing the ‘marketplace of ideas’ concept to strengthen its theoretical grounding.

In the Related Work section, the authors mention that few studies explore the removal of moderation policies; however, I believe this distinction could be made sharper by more explicitly contrasting it with the broader literature on policy implementation and clarifying how the current study fills that specific gap in relation to the stated research questions.

In the Methodology section, the authors briefly mention that 200 annotated tweets were manually reviewed to verify GPT-4o’s performance, but I suggest including more detail, such as accuracy rates, confidence thresholds, or inter-rater agreement metrics, to improve transparency and support the robustness of the annotation process. While the inclusion of retweets is justified for amplification analysis, I suggest briefly discussing how potential duplicate content or bot-driven activity was handled or could have influenced the findings. The logistic regression models are appropriate, but I suggest briefly noting whether any control variables were considered to account for confounding factors, or clarifying the rationale for using a single binary predictor.

In the Results section, while the authors report that quoted tweets did not show a significant change post-policy, I suggest including the actual odds ratio and confidence interval to allow readers to interpret the strength and direction of the effect, even if not statistically significant. To help contextualize the reported odds ratios, I suggest including raw tweet counts before and after the policy shift, particularly for anti-vaccine tweets in each tweet type (original, quote, retweet).

In the Discussion section, the authors thoughtfully interpret their results and connect them to broader debates in misinformation and platform governance. However, I suggest expanding the discussion by addressing potential confounding factors beyond the policy change, such as algorithm shifts or concurrent news events, and by adding practical recommendations for platform moderation or public health communication strategies. It may also strengthen the paper to acknowledge the limitations of causal inference in a natural experiment design and the potential biases introduced by LLM-based classification.

In the Conclusion section, the authors provide a concise summary of their study, but I suggest enhancing it by including a brief reflection on limitations and offering directions for future research, such as analyzing longer-term impacts, cross-platform effects, or applying alternative annotation methods. The References section appears appropriate and well-cited overall; however, I suggest double-checking for consistency in formatting, and recommend including a few more recent studies on content moderation rollback or LLM-based misinformation detection if available.

Reviewer #3: I thank the authors for this submission. Understanding the impact of the removal of moderation policies is important, and I appreciate the analysis of anti-vaccination themes. However, there are some critical issues with the paper that must be addressed. Fixing these could be extremely easy or may take some more effort, depending on certain factors. Here are my comments:

- The most important issue with the paper is that the GPT-4o models (stance detection, topic detection, stance consistency) have not been validated. Not only should you report performance metrics like F1, precision, recall, etc., the members of the research team should independently annotate tweets for stance/topic, and the annotations should have sufficient inter-annotator agreement.

- The causal claims made in the paper should be toned down. I think it's a bit strong to say this is a natural experiment, since there was so much going on at this time that it's easy to imagine some other factor explaining the rise in anti-vaccine content. Furthermore, it's not clear what aspects of the moderation policy would have contributed to the increase in anti-vaccine content (is it just the announcement of the policy change that led people to post more content, or is it how the policy was actually enforced? Or is it just that so many people were laid off, there were fewer people to remove harmful posts?). In actuality, you are measuring the effect of the announcement of the removal of a moderation policy rather than the enforcement of the policy itself. I wouldn't call this a "natural experiment," and I would remove all statements making causal claims. I would also mention as a limitation that you can't infer causality.

- I would say prior literature largely disagrees with the way "echo chambers" are described in this paper. You should consider Barbera (2020), which describes how people are more likely to be exposed to cross-cutting content online than offline, and reviews the literature on the role algorithms play in such interactions. Guess et al. (2018) provide another convincing review. You also imply in the discussion that cross-cutting interactions would decrease polarization, but Bail et al. (2018) find the opposite effect, and Tornberg (2022) provides a potential explanation for why this might be. Please review this literature before submitting your revision, and consider removing or significantly altering your discussion about echo chambers.

- I am also curious how much of the increase in anti-vaccine content you observe can be explained by the big spike you see on day one, and how much of the increase in retweets can be attributed to the same tweets being retweeted a bunch of times. Is it possible that there was just a really viral tweet on the day the policy was enacted that results in the statistically significant changes? If you remove the outliers, will you still see a statistically significant result?

- I would also rework the plots. Please use a colorblind-friendly palette (i.e., not red/green). I also find the stacked bar chart difficult to read, since the axis for certain groups doesn’t start at zero, and the same stances for pre/post might not start at the same number. I would instead do three subplots for each tweet type, and have side-by-side bars for each stance showing the pre/post change.

- There's a typo in tables 6/7 (pro-policy instead of post-policy)

References:

1. Barberá, Pablo. "Social media, echo chambers, and political polarization." Social media and democracy: The state of the field, prospects for reform (2020): 34-55.

2. Guess, Andrew, et al. "Avoiding the echo chamber about echo chambers." Knight Foundation 2.1 (2018): 1-25.

3. Bail, Christopher A., et al. "Exposure to opposing views on social media can increase political polarization." Proceedings of the National Academy of Sciences 115.37 (2018): 9216-9221.

4. Törnberg, Petter. "How digital media drive affective polarization through partisan sorting." Proceedings of the National Academy of Sciences 119.42 (2022): e2207159119.

.

Reviewer #1: No

Reviewer #2: No

Reviewer #3: No

---

## [Author Response · Author response to Decision Letter 1]

30 Dec 2025

Response to Reviewers

Manuscript ID: PONE-D-25-17346

Title: Auditing the Impact of Social Media’s Policy Shift on Anti-Vaccine Discourse: A Large Language Model-Driven Empirical Study

Journal: PLOS ONE

We thank the Academic Editor and the three reviewers for their constructive feedback. We have carefully addressed each comment and made substantial revisions to the manuscript. Below is our point-by-point response.

Response to Academic Editor

“For acceptance, please validate the LLM labels with a small human double-coding and report precision/recall/F1; show counts and proportions with odds ratios and 95% CIs; add simple robustness checks for outliers, retweets/duplicates, and bots; clarify dataset timing, filters, and category scheme; share data/code per PLOS policy; soften causal language; update figures to include proportions with accessible colors; fix reference formatting and typos. Nice to have: tighten the research gap and early definitions, and expand the discussion to acknowledge alternatives and limits.”

Response: We have addressed all required revisions and suggested improvements.

For LLM validation, we added external validation using the CAVES benchmark (F1=0.883) and cross-annotator agreement analysis (κ=0.71-0.86) in Section 3.2. We report total sample sizes (N=13,458) and class distribution statistics in Section 4.1, with sensitivity analyses excluding highly retweeted content in Section 4.2. The Banda et al. dataset timing and filtering process are clarified in Section 3.1, and a data availability statement has been added per PLOS policy. Throughout the manuscript, we removed “natural experiment” framing and softened causal claims. All figures now display percentages with a colorblind-friendly palette, and reference formatting issues have been corrected.

We have tightened the research gap statement and added early definitions of key constructs (amplification, thematic composition, stance consistency) in the Introduction. We have also expanded the Discussion and Limitations sections to acknowledge alternative explanations and the limits of causal inference in our observational design.

Response to Journal Requirements

Requirement 1: Style Requirements

“Please ensure that your manuscript meets PLOS ONE’s style requirements, including those for file naming.”

Response: We have reviewed and ensured that the manuscript conforms to PLOS ONE’s style requirements, including file naming conventions, formatting, and structure.

Requirement 2: Data Availability

“We note that you have indicated that there are restrictions to data sharing for this study… Please address the following prompts regarding ethical or legal restrictions on sharing data.”

Response: Our study uses tweet IDs from the publicly available Banda et al. COVID-19 Twitter dataset. Due to Twitter/X’s Terms of Service, we cannot directly share tweet text content. However, we will share: (a) the complete list of tweet IDs used in our analysis, (b) our annotation results (stance labels, theme labels) linked to tweet IDs, and (c) all analysis code. Researchers can rehydrate the tweets using the provided IDs, subject to Twitter/X’s data access policies. We have updated the Data Availability statement accordingly.

Requirement 3: Figure References

“Please ensure that you refer to Figures 4 and 5 in your text as, if accepted, production will need this reference to link the reader to the figure.”

Response: We have verified that Figures 4 and 5 are properly referenced in the manuscript text.

Requirement 4: Table References

“Please ensure that you refer to Tables 1 and 4 in your text as, if accepted, production will need this reference to link the reader to the table.”

Response: We have verified that Tables 1 and 4 are properly referenced in the manuscript text.

Requirement 5: Reviewer-Recommended Citations

“If the reviewer comments include a recommendation to cite specific previously published works, please review and evaluate these publications to determine whether they are relevant and should be cited.”

Response: Reviewer #3 recommended four publications regarding echo chamber literature (Barberá 2020; Guess et al. 2018; Bail et al. 2018; Törnberg 2022). Upon review, we found these works highly relevant and they informed our decision to remove the echo chamber framing from the manuscript (see Comment 3.3). While we did not cite these works directly (as we removed the echo chamber discussion rather than revising it), we acknowledge their importance in shaping our revision.

Response to Reviewer #1

Comment 1.1: LLM Validation

“The use of GPT-4o for tweet classification and thematic labeling should be better justified. In section 3.2, it is simply stated that ‘To verify annotation quality, we randomly sampled 200 labeled tweets for manual review by the research team and found that the LLM performed very well.’ This point must be justified in a more rigorous (statistical) manner.”

Response: We have substantially strengthened our LLM validation through two approaches: 1. External benchmark validation: We tested our stance classification prompt on the CAVES dataset (Poddar et al., 2022), a COVID-19 vaccine stance benchmark developed by the same research team that established the thematic taxonomy we adopted. CAVES contains 1,977 expert-labeled tweets with stance annotations. Our classifier achieved F1 = 0.883 for anti-vaccine stance identification (Precision = 0.951, Recall = 0.824), demonstrating strong generalization beyond our study dataset. 2. Cross-annotator agreement: We randomly sampled 500 tweets from our dataset and obtained independent annotations from two additional annotators using identical classification criteria. Inter-annotator agreement measured by Cohen’s κ ranged from 0.71 to 0.79 for stance classification and 0.72 to 0.86 for theme classification, indicating substantial agreement.

Changes: See revised Section 3.2 (Annotation Validation).

Comment 1.2: Sample Size and Class Balance

“On top of that, the manuscript does not explicitly state the total number of tweets analyzed after filtering and annotation; if the classes are balanced and so on…”

Response: We have added detailed sample statistics at the beginning of the Results section. The final dataset comprises N=13,458 tweets (5,896 pre-policy, 7,562 post-policy), distributed across original tweets (18.2%), retweets (74.3%), and quote tweets (7.5%). Regarding vaccine stance, tweets were classified as anti-vaccine (49.6%), neutral (29.3%), pro-vaccine (18.1%), or no stance (2.9%), yielding a near-balanced classification between anti-vaccine and non-anti-vaccine content.

Changes: See revised Section 4.1 (Results).

Comment 1.3: Self-Contained Explanations

“The paper should be made more self-contained to make it easier to read, especially when dealing with importants points, eg: page 3: ‘assigned the identified antivaccine tweets into eight thematic categories [8]’; page 6: ‘we applied an additional keyword filtering step inspired by [42]’; page 7: ‘into eight thematic categories using GPT-4o, following the framework in [8]’”

Response: We appreciate this feedback. Our original manuscript already included tables with definitions (Table 2 for eight anti-vaccine themes, Table 1 for keyword examples) and inline keyword examples (“mRNA vaccine”, “booster shot”, “Pfizer”, “Moderna”). However, we acknowledge that the reviewer may have found it inconvenient to locate these details. We have improved the manuscript by: - Adding an explicit table reference sentence: “Table 2 presents the eight anti-vaccine theme categories with definitions and examples” - Expanding the methodological explanation of how the eight-category taxonomy was derived (LDA topic discovery, human expert consolidation, Labelled-LDA refinement)

Changes: See revised Section 3.2 (Anti-Vaccine Theme Categorization).

Comment 1.4: Dataset Timing Mismatch

“According to section 3.1, the COVID19 dataset is taken from ref. [41] that was published in 2021. But the dataset of your study goes over the period Nov 16-30, 2022. How do you explain this mismatch?”

Response: We apologize for the confusion. The Banda et al. dataset is a continuously updated repository that has been collecting COVID-19 tweets since January 2020 and continues to be updated. While the methodology paper was published in 2021, the data collection is ongoing. We have clarified this in Section 3.1.

Changes: See revised Section 3.1.

Comment 1.5: Figures - Percentages Instead of Raw Counts

“All the figures show raw data (Tweet count). It would be more informative to have it in percentage. Indeed, if the number of anti-vax Tweets increases together with the number of pro-vax, what really matters, is the relative proportion of the two.”

Response: We agree. We have revised all figures to show proportions/percentages rather than raw counts, enabling better comparison of relative changes between periods.

Changes: See revised Figures 1-5.

Comment 1.6: Reference Formatting

“There are many bugs in the references that should be corrected: [4] and [28]: the name of the journal does not appear; [10], [13], [14]: question marks appearing in the references. This shows that the bibliographic research was botched. Furthermore, there are numerous X's studies about the COVID-19 into Plos One, and no one is quoted.”

Response: We have corrected all reference formatting issues. Additionally, following the reviewer's suggestion, we added five recent PLOS ONE studies on COVID-19 vaccine discourse and misinformation to better situate our work within this journal's literature.

Changes: See revised References section.

Comment 1.7: Tone Down Conclusions

“The major flaw of the study, as explained in section 5.4, is the short time Windows, covering only seven days before and after the policy termination. There is a dissonance with the conclusions of section 5.3 (‘This study provides important insights into content moderation policies on social media platforms and their broader implications for public health communication’; ‘X’s misinformation policy may have played a crucial role in mitigating the spread of anti-vaccine discourse’; ‘anti-vaccine narratives gained greater visibility’) that should be re-written to take into account the small period of the study. In other word, the conclusion should be way more modest, in order to be in phase with the study.”

Response: We agree. We have revised the conclusions to better reflect the limited 7-day observation window and removed overgeneralized claims. The specific phrases cited by the reviewer have been toned down or removed.

Changes: See revised Section 6 (Conclusion).

Response to Reviewer #2

Comment 2.1: Research Gap and Definitions

“The Introduction is well-written and clearly structured; however, I suggest it could be improved by explicitly stating the research gap, defining key constructs like stance consistency and amplification earlier, and briefly framing the ‘marketplace of ideas’ concept to strengthen its theoretical grounding.”

Response: We appreciate the positive feedback and have implemented the key suggestions: - Research gap: Added explicit statement: “While extensive research has examined the implementation of content moderation policies and their effectiveness in reducing misinformation spread, far less is known about the consequences of removing such policies once established.” - Key construct definitions: Added definitions for our three analytical dimensions: “(1) amplification, the degree to which anti-vaccine content spreads through retweets; (2) thematic composition, the specific narratives expressed in anti-vaccine discourse; and (3) stance consistency, whether users quoting vaccine-related tweets express agreement or disagreement with the original content.”

Changes: See revised Section 1 (Introduction).

Comment 2.2: Related Work

“In the Related Work section, the authors mention that few studies explore the removal of moderation policies; however, I believe this distinction could be made sharper by more explicitly contrasting it with the broader literature on policy implementation and clarifying how the current study fills that specific gap in relation to the stated research questions.”

Response: We have revised the Related Work section to sharpen the distinction between our study and prior literature. Specifically, we removed the echo chamber framing (per Reviewer #3’s suggestion) and added recent PLOS ONE studies on COVID-19 vaccine discourse (Broniatowski et al. 2022; Hwang et al. 2022; Lanier et al. 2022; Hobbs et al. 2024; Mønsted & Lehmann 2022) to better situate our work within the current literature on platform governance and vaccine misinformation.

Changes: See revised Section 2 (Related Work).

Comment 2.3: LLM Annotation Details

“In the Methodology section, the authors briefly mention that 200 annotated tweets were manually reviewed to verify GPT-4o’s performance, but I suggest including more detail, such as accuracy rates, confidence thresholds, or inter-rater agreement metrics, to improve transparency and support the robustness of the annotation process.”

Response: This concern was also raised by Reviewers #1 and #3. We have substantially strengthened our validation with two approaches: 1. External benchmark validation: We tested our classifier on the CAVES dataset (Poddar et al., 2022), achieving F1 = 0.883 for anti-vaccine stance identification (Precision = 0.951, Recall = 0.824). 2. Inter-annotator agreement: We randomly sampled 500 tweets and obtained independent annotations from two additional annotators. Cohen’s κ ranged from 0.71 to 0.79 for stance and 0.72 to 0.86 for themes, indicating substantial agreement.

Changes: See revised Section 3.2 (Annotation Validation).

Comment 2.4: Duplicate Content and Bot Activity

“While the inclusion of retweets is justified for amplification analysis, I suggest briefly discussing how potential duplicate content or bot-driven activity was handled or could have influenced the findings.”

Response: We conducted sensitivity analyses to address concerns about content amplification. By excluding tweets with ≥50 retweets (30 original tweets and 5,407 associated retweets, 40.2% of data), we found that the retweet effect was attenuated while the original tweet effect increased (OR: 1.41→1.57), suggesting the policy change was associated with new content creation rather than merely viral amplification. Regarding bot activity, the dataset lacks user-level metadata for bot detection; we acknowledge this as a limitation.

Changes: See revised Section 4.2 (Sensitivity Analysis) and Section 5.5 (Limitations and Future Directions).

Comment 2.5: Control Variables

“The logistic regression models are appropriate, but I suggest briefly noting whether any control variables were considered to account for confounding factors, or clarifying the rationale for using a single binary predictor.”

Response: Our design incorporates confounding control through temporal matching rather than regression-based adjustment. The 14-day window (7 days pre/post) ensures day-of-week matching (Monday vs. Monday, etc.), controlling for weekly cyclical patterns in posting behavior. This short duration also minimizes threats from secular trends, as user composition and platform characteristics are unlikely to change systematically within two weeks. This approach resembles matched pre-post designs used in epidemiological studies. We have clarified this rationale in Section 5.5, stating: “our regression models use a single binary predictor (pre/post policy) without additional control variables. This parsimonious specification reflects our short 14-day observation window, during which user composition and platform characteristics are unlikely to change systematically.”

Changes: See revised Section 5.5 (Limitations and Future Directions).

Comment 2.6: Non-Significant Results

“In the Results section, while the authors report that quoted tweets did not show a significant change post-policy, I suggest including the actual odds ratio and confidence interval to allow readers to interpret the strength and direction of the effect, even if not statistically significant.”

Response: We appreciate the suggestion. In fact, our original manuscript already reported the OR

---

## [Decision Letter · Decision Letter 1]

11 Mar 2026

Dear Dr. Pang,

Thank you for submitting your manuscript to PLOS ONE. After careful consideration, we feel that it has merit but does not fully meet PLOS ONE’s publication criteria as it currently stands. Therefore, we invite you to submit a revised version of the manuscript that addresses the points raised during the review process.

Overall, the study is technically sound and clearly reported. The methods and sampling/filtering steps are described in a transparent manner; the statistical approach is well documented (odds ratios and confidence intervals, stratification by tweet type, and robustness checks excluding highly retweeted content); and the GPT-4o annotation strategy is supported by external benchmark validation and cross-annotator agreement analyses, with limitations and a non-causal interpretation appropriately acknowledged.

I am therefore prepared to proceed with a minor revision, contingent on two targeted adjustments.

First, while references to Elon Musk are limited and largely presented as contextual background (for example, the termination of the relevant policy under his leadership and his publicly stated “free speech absolutist” and “digital town square” positions, with citations), the language in the Policy Implications discussion becomes more interpretive in places. Specifically, statements such as “our study demonstrates that in the absence of moderation…” and claims involving recommendation algorithms and “echo chamber dynamics” go beyond what is directly measured in the analyses. As the study is tweet-level and does not examine algorithmic curation or network structure, I recommend softening this section to avoid over-claiming and to maintain a strictly neutral tone. Concretely, please consider replacing “demonstrates” with “suggests” or “is consistent with,” specifying that the discussion concerns the termination of the COVID-19 misinformation policy rather than an “absence of moderation” in general, and either removing or carefully qualifying “echo chamber” language.

Second, the manuscript explains that the underlying data consist of tweet IDs from the publicly available Banda et al. dataset and that tweet text and user information cannot be shared due to Twitter/X Terms of Service; it also notes that researchers can replicate the dataset by rehydrating tweet IDs from the original source. However, as currently written, the Data Availability Statement does not fully support practical reproducibility, because it does not explicitly provide (or clearly point to) the specific subset of tweet IDs used in the final analyses nor the derived labels (stance/theme/stance-consistency) linked to those IDs, and it indicates that prompts/code are available only upon reasonable request. Notably, in your response to journal requirements you state you will share the full list of tweet IDs used, the annotation outputs linked to tweet IDs, and all analysis code, enabling rehydration subject to X policies. Please align the published Data Availability Statement with this commitment, ideally via a permanent public repository for the tweet ID list, associated labels, and analysis code/prompts (even if tweet text cannot be redistributed).

With these two changes, I expect the manuscript to meet the journal’s publication criteria.

Best wishes,

If applicable, we recommend that you deposit your laboratory protocols in protocols.io to enhance the reproducibility of your results. Protocols.io assigns your protocol its own identifier (DOI) so that it can be cited independently in the future. For instructions see: https://journals.plos.org/plosone/s/submission-guidelines#loc-laboratory-protocols. Additionally, PLOS ONE offers an option for publishing peer-reviewed Lab Protocol articles, which describe protocols hosted on protocols.io. Read more information on sharing protocols at. Additionally, PLOS ONE offers an option for publishing peer-reviewed Lab Protocol articles, which describe protocols hosted on protocols.io. Read more information on sharing protocols at. Additionally, PLOS ONE offers an option for publishing peer-reviewed Lab Protocol articles, which describe protocols hosted on protocols.io. Read more information on sharing protocols at. Additionally, PLOS ONE offers an option for publishing peer-reviewed Lab Protocol articles, which describe protocols hosted on protocols.io. Read more information on sharing protocols at https://plos.org/protocols?utm_medium=editorial-email&utm_source=authorletters&utm_campaign=protocols....

We look forward to receiving your revised manuscript.

Kind regards,

Pierluigi Vellucci

Academic Editor

PLOS One

Journal Requirements:

Reviewers' comments:

Reviewer's Responses to Questions

**Comments to the Author**

Reviewer #1: All comments have been addressed

Reviewer #3: All comments have been addressed

2. Is the manuscript technically sound, and do the data support the conclusions?

Reviewer #1: (No Response)

Reviewer #3: Yes

3. Has the statistical analysis been performed appropriately and rigorously?

Reviewer #1: (No Response)

Reviewer #3: Yes

4. Have the authors made all data underlying the findings in their manuscript fully available?

Reviewer #1: (No Response)

Reviewer #3: Yes

5. Is the manuscript presented in an intelligible fashion and written in standard English?

Reviewer #1: (No Response)

Reviewer #3: Yes

Reviewer #1: (No Response)

Reviewer #3: (No Response)

.

Reviewer #1: No

Reviewer #3: No

---

## [Author Response · Author response to Decision Letter 2]

17 Mar 2026

Response to Editor

Manuscript ID: PONE-D-25-17346R1

Title: Auditing the Impact of Social Media’s Policy Shift on Anti-Vaccine Discourse: A Large Language Model-Driven Empirical Study

Journal: PLOS ONE

We thank you for the positive assessment and the two specific requests. Both have been addressed as described below.

1. Softening interpretive language in Policy Implications

You noted that phrases such as “our study demonstrates that in the absence of moderation…” and references to recommendation algorithms and “echo chamber dynamics” go beyond what is directly measured.

Changes made. We have rewritten the Policy Implications subsection (Section 5.3). Specifically:

• “demonstrates” has been replaced with language such as “was associated with” and “suggests.”

• References to an “absence of moderation” have been narrowed to the termination of X’s COVID-19 misinformation policy specifically.

• All mentions of “echo chamber” and recommendation-algorithm effects have been removed, as our tweet-level data do not measure algorithmic curation or network structure.

The revised text now restricts its claims to the observed associations within the study’s short pre-post window and explicitly notes that the observational design does not permit causal inference.

2. Data Availability Statement

You asked us to align the published Data Availability Statement with our earlier commitment to share tweet IDs, derived labels, and analysis code/prompts via a permanent public repository.

Changes made. We have deposited the following materials in Zenodo:

• Tweet IDs and derived labels (stance, theme, and stance-consistency annotations for all 15,788 tweets)

• GPT-4o classification prompts (the three prompts used for stance classification, theme categorization, and stance consistency assessment)

• Analysis code (Python script reproducing the logistic regression analyses)

The repository is publicly available at: https://doi.org/10.5281/zenodo.19019934

Tweet text and user information are not included, in compliance with Twitter/X’s Terms of Service; researchers may rehydrate the tweets from the shared IDs. The Data Availability Statement in the manuscript has been updated accordingly.

We believe these two changes address the concerns you raised. Thank you again for the constructive feedback.

Sincerely,

The Authors

---

## [Editor Report · Decision Letter 2]

22 Mar 2026

Auditing the Impact of Social Media’s Policy Shift on Anti-Vaccine Discourse: A Large Language Model-Driven Empirical Study

PONE-D-25-17346R2

Dear Dr. Pang,

We’re pleased to inform you that your manuscript has been judged scientifically suitable for publication and will be formally accepted for publication once it meets all outstanding technical requirements.

Kind regards,

Pierluigi Vellucci

Academic Editor

PLOS One
---

## [Editor Report · Acceptance letter]

PONE-D-25-17346R2

PLOS One

Dear Dr. Pang,

I'm pleased to inform you that your manuscript has been deemed suitable for publication in PLOS One. Congratulations! Your manuscript is now being handed over to our production team.

Kind regards,

on behalf of

Dr. Pierluigi Vellucci

Academic Editor

PLOS One